# Roles of Oxidative Stress in Acute Tendon Injury and Degenerative Tendinopathy—A Target for Intervention

**DOI:** 10.3390/ijms23073571

**Published:** 2022-03-25

**Authors:** Pauline Po Yee Lui, Xing Zhang, Shiyi Yao, Haonan Sun, Caihao Huang

**Affiliations:** 1Department of Orthopaedics and Traumatology, The Chinese University of Hong Kong, Hong Kong 999077, China; shiyiyao@link.cuhk.edu.hk; 2Institute of Metal Research, Chinese Academy of Sciences, Shenyang 110866, China; xingzhang@imr.ac.cn (X.Z.); hnsun21b@imr.ac.cn (H.S.); chhuang17s@imr.ac.cn (C.H.)

**Keywords:** tendon injury, tendinopathy, oxidative stress, reactive oxygen species

## Abstract

Both acute and chronic tendon injuries are disabling sports medicine problems with no effective treatment at present. Sustained oxidative stress has been suggested as the major factor contributing to fibrosis and adhesion after acute tendon injury as well as pathological changes of degenerative tendinopathy. Numerous in vitro and in vivo studies have shown that the inhibition of oxidative stress can promote the tenogenic differentiation of tendon stem/progenitor cells, reduce tissue fibrosis and augment tendon repair. This review aims to systematically review the literature and summarize the clinical and pre-clinical evidence about the potential relationship of oxidative stress and tendon disorders. The literature in PubMed was searched using appropriate keywords. A total of 81 original pre-clinical and clinical articles directly related to the effects of oxidative stress and the activators or inhibitors of oxidative stress on the tendon were reviewed and included in this review article. The potential sources and mechanisms of oxidative stress in these debilitating tendon disorders is summarized. The anti-oxidative therapies that have been examined in the clinical and pre-clinical settings to reduce tendon fibrosis and adhesion or promote healing in tendinopathy are reviewed. The future research direction is also discussed.

## 1. Tendon Fibrosis, Adhesion and Degenerative Tendinopathy—Unresolved Sports Medicine Problems

Tendons and ligaments are subject to high tensile loads and are easily torn as a result of overuse or trauma, resulting in significant pain and disability. Tendons and ligaments heal poorly after injury, posing a significant burden to society. Tendon and ligament injuries account for 30% of all musculoskeletal consultations [1]. More than 32 million acute and chronic tendon and ligament injuries occur annually in the United States [2]. The outcomes of both conservative treatments and surgical repair of tendon are not satisfactory, with long healing time, scar tissue formation, adhesion, occasional bone formation and high re-rupture rate. 

Tendon scarring and adhesion are a fibrotic tissue response secondary to an inflammatory process. Persistent inflammation aggravates fibrosis [3]. After tendon injury, macrophages are recruited to the healing site. They produce a plethora of inflammatory cytokines that induce inflammatory response, leading to the activation and recruitment of other immune cells, tendon stem/progenitors and tenocytes. Several studies have indicated that macrophages negatively regulate tendon healing. Macrophage depletion was reported to reduce cell proliferation and extracellular matrix accumulation, reduce heterotopic ossification, improve the ultimate tensile strength and alleviate peritendinous fibrosis of injured tendons [4,5,6].

The accumulation of microinjuries during loading as a result of unsuccessful tendon regeneration after acute injuries is a predisposing factor for the development of chronic degenerative tendinopathy. Degenerative tendinopathy is a broad spectrum of chronic tendon disorders presented with local pain, stiffness, physical disability and ultimately tendon rupture due to tendon overuse and aging. Approximately 30% of adults over the age of 60 have a rotator cuff tear, and it increases to 62% in adults over the age of 80. After 66 years of age, there is a 50% likelihood of having bilateral rotator cuff tears. Tendon overuse initiates an inflammatory cascade that promotes aberrant tendon-derived stem cell (TDSC) differentiation, causing tissue metaplasia and failed tenogenic differentiation. Indeed, the tenogenic properties of tendon-derived stem cells (TDSCs) are compromised in an inflammatory environment [7,8,9]. There are currently no effective treatments for tendon fibrosis, adhesion and degenerative tendinopathy, largely due to our limited understanding of the tendon healing and failed healing processes. Better understanding of the disease etiopathogenesis is required to develop effective tendon regenerative approaches.

## 2. Oxidative Stress in the Pathogenesis of Tendon Fibrosis, Adhesion and Degenerative Tendinopathy

Reactive oxygen species (ROS) are continually generated during normal cell metabolism. Many external factors, such as trauma, environmental and physiological stimuli, can enhance ROS production. Mechano-sensitive tissues, such as tendons, are continuously exposed to oxidative stress during exercise. ROS can damage lipid, proteins and DNA in cells and tissues. Sustained oxidative stress is a major contributing factor to tendon fibrosis, adhesion, and pathological changes in tendinopathy. However, the sources and mechanisms of oxidative stress in tendon fibrosis and degenerative tendinopathy are not precisely known. Many factors, such as mechanical overuse, ischemia-refusion injury and hyperthermia in exercising tendon, release of inflammatory cytokines and growth factors after tendon injury, drug exposure such as fluoroquinolones, anesthetics, steroid, nonsteroidal anti-inflammatory drugs (NSAIDs) and metabolic factors, including hyperglycemia, hyperlipidemia and hypercholesterolemia, can prompt oxidative stress in tendon tissue, triggering cellular apoptosis, tendon fibrosis, and tendon degeneration.

This study aims to appraise the clinical and pre-clinical evidence regarding the negative impacts of oxidative stress on tendon. 

Literature in PubMed was searched and screened with the keywords ((tendon [Title/Abstract]) AND ((ROS[Title/Abstract]) OR (reactive oxygen species [Title/Abstract]) OR (oxidative stress [Title/Abstract]))) on 23 February 2022. In total, 123 titles were retrieved. An additional 21 articles titles were retrieved from the reference list of the articles. All original clinical and pre-clinical research articles (81 articles), directly related to the effects of oxidative stress and activators or inhibitors of oxidative stress on tendon, were reviewed and included in this review article. 

## 3. Clinical and Pre-Clinical Evidence of Potential Sources and Mechanisms of Oxidative Stress in Tendon

### 3.1. Clinical Evidence

The tendon constantly generates ROS after exercise and traumatic injury [10]. Excessive ROS exceeding the tendon’s antioxidant capacity can cause inflammation, resulting in tendon damage. An early clinical study reported an increased expression of peroxiredoxin 5, a thioredoxin peroxidase with antioxidant properties, in fibroblasts and endothelial cells in tendinopathic tendon [11], which protected human tendon cells from apoptosis and loss of cellular function during oxidative stress in vitro [12]. Two recent clinical studies showed that the level of superoxide-induced oxidative stress increased and was associated with degeneration of rotator cuff and recurrent tear after arthroscopic rotator cuff repair [13,14]. In addition, the ROS level in the synovial fluid of patients with painful degenerative rotator cuff tears and osteoarthritis was shown to be higher, compared to that in the control group [15]. In another study, proteomic analysis also revealed that the synovial tissue of patients with rotator cuff tear and persistent shoulder pain showed significantly higher expression of S100A11 (a protein involved in inflammatory response), PLIN4 (a protein for lipid droplet formation), and HYOU1 (an endoplasmic reticulum protein that accumulates under hypoxia to protect cells from hypoxia and hence is involved in oxidative stress response), compared to the expression in the control patients, confirming that rotator cuff tendinopathy is closely related to oxidative stress and inflammation [16]. One research group measured the serum concentration of hydroperoxides of organic compounds as an oxidative stress-related marker and reported that the level of oxidative stress was significantly higher in elite professional soccer showing ultrasonographic (US) features of tendon damage compared to those with normal US pictures, even after adjustment for age and body mass index, suggesting that oxidative stress favors tendon damage [17].

### 3.2. Pre-Clinical Evidence

#### 3.2.1. Tendon Injury

Both acute and chronic tendon injuries can induce oxidative stress and inflammation in tendon. Oxidative stress and pro-inflammatory cytokines show a complex cross talk, impacting cell viability, cell fate and extracellular matrix (ECM) degradation. Pro-inflammatory cytokines are upregulated by oxidative stress, which may activate apoptosis and matrix degeneration in tendon. Some cytokines, such as IL-β and TNF-α, further induce ROS production through nicotinamide adenine dinucleotide phosphate oxidase (NOX), in tendon cells [18].

Tendon fibrosis and tendon adhesion are two major issues encountered after acute tendon injury. Proteomic analysis of adult and fetal tendons after injury showed that there was a significant up-regulation of oxidative stress, pro-inflammatory factors, matrix degrading enzymes and activation of neutrophils in the adult tendon defects, which were not observed in fetal tendon healing, supporting the association of oxidative stress and inflammation with poor healing outcome in adult tendon healing [19]. The ROS level was shown to increase significantly in the injured tendon, compared to the normal tendon in a rat patellar tendon window injury model [20]. In addition to acute tendon injury, oxidative stress and inflammation are also associated with tendinopathy. Mice deficient of an antioxidant enzyme, superoxide dismutase 1 (Sod1), showed degeneration in the supraspinatus tendon entheses, supporting that intracellular oxidative stress contributes to rotator cuff degeneration in tendinopathy [21]. In another study, oxidative stress, inflammation, apoptosis, and autophagy were triggered in the collagenase-induced Achilles tendinopathy rat model [22]. 

Overproduction of hydrogen peroxide (H_2_O_2_), an important oxidative stress inducer, caused oxidative damage, activation of endoplasmic reticulum (ER) stress and apoptotic death of tenocytes [23]. Treatment of human tenocytes with H_2_O_2_ induced apoptosis with cytochrome c release from mitochondria and caspase-3 activation [24]. N-acetylcysteine (NAC), a cysteine donor and glutathione precursor, was shown to protect rotator cuff fibroblasts from ROS-mediated hypoxic cell apoptosis [25] and silver nanoparticle-triggered ROS-induced apoptosis of TDSCs [26]. In addition to cell apoptosis, H_2_O_2_ also induced the mRNA expression of matrix metalloproteinase-1 (MMP1) and protein expression of pro-MMP1 in human tendon cells, suggesting that oxidative stress plays an important role in tendon matrix degradation [27]. Besides tenocytes, oxidative stress can also reduce the viability, self-renewal and multi-lineage differentiation potential of TDSCs, thereby reducing their regenerative and tenogenic function. In this regard, H_2_O_2_ was reported to impair colony formation, stemness marker expression, and differentiation capacity as well as showing suppressed cell migration, cell viability, apoptosis, and proliferation of TDSCs [28,29,30]. In a rat patellar tendon window injury model, subcutaneous injection of H_2_O_2_ over the tendon was demonstrated to induce more tendon swelling, pain-associated gait asymmetry, disorganized fibrous tissue formation and hypoechogenic areas in tendon tissue, compared to the saline-treated group [31].

Peritendinous adhesion following tendon injuries prevents tendon excursion, and eventually results in poor function. As a mediator of tissue fibrosis, ROS plays an important role in tendon adhesion. For example, tendon transection and repair were reported to induce tendon adhesion in rats, with reduced superoxide dismutase (SOD) activity and glutathione (GSH) contents as well as increased malondialdehyde (MDA) level (a lipid peroxidation marker) [32]. A similar result was observed in another study in which the expression of oxidative stress markers and inflammatory cytokines increased within 24 h after Achilles tendon injury and was associated with peritendinous adhesion formation in a rat model [33]. Tendon adhesion is common after flexor tendon injury and repair. In a chicken flexor tendon injury model, the tissue level of GSH decreased while the level of glutathione disulfide (GSSH) increased post-injury, along with peritendinous adhesion [34]. The administration of an antioxidant reduced tendon adhesion, further confirming the pathogenic role of oxidative stress in tendon adhesion [34].

#### 3.2.2. Overuse

Tendon loading is a primal factor of cellular stress in tenocytes. Microarray analysis showed that genes related to ROS, in addition to genes related to inflammation and angiogenesis, were strongly regulated by mechanical loading after Achilles tendon injury in rats [35]. Tendon overuse increased the plasma total oxidant status (TAS) and vascular endothelial growth factor (VEGF) in rats compared to the cage activity group [36]. The unloading of tendon fascicles may occur due to overloading and the subsequent rupture of sub-tissue structures. The exposure of unloaded tendon fascicles to supra-physiological oxygen and temperature levels generated ROS, induced MMPs, and activated immune system, casing ECM degeneration and functional impairment of the tendon explants, which could be rescued by inhibitors of ROS and MMPs [37].

A recent study generated an interesting hypothesis about how collagen mechanoradicals might generate oxidative stress in tendon [38]. The authors showed that collagen in tendon tissue acted as a mechanoradical sink against mechano-oxidative damage. The mechanical stress on collagen generated radicals in the vicinity of the collagen crosslinks. The radicals then migrated to the adjacent clusters of romantic residues and stabilized as dihydroxyphenylalaniine (DOPA) arising from the oxidation of conserved tyrosine or phenylalanine residues around the collagen crosslinks. The DOPA was finally converted into H_2_O_2_ in the presence of water [38]. This is consistent with a previous study, which has shown that tendon collagen contains cryptic bioactive peptides C2 and E1 that, when exposed or liberated, more effectively scavenged hydroxyl radicals and hence conferred better protection to cells against oxidative stress, compared to undigested collagen [39]. Combination of coated C2 and dissolved E1 showed faster gap closure of Vero cells, compared to undigested collagen coating or no coating, with or without H_2_O_2_ exposure, in a scratch assay [39].

#### 3.2.3. mTOR Pathway

Mechanistic target of rapamycin (mTOR), the catalytic subunit of two distinct protein complexes, mTORC1 and mTORC2, regulates many metabolic processes, including protein synthesis, cell survival and proliferation. It has a dual role in tendons. On one hand, mTOR knockout caused tendon defects in mice and a high concentration of rapamycin, an mTOR inhibitor, impaired proliferation and tenogenesis of TDSCs [40]. The mRNA expression of mTOR was suppressed in human tendinopathy samples, compared to normal tendon tissues, suggesting that deficiency of mTOR might be related to the pathogenesis of tendinopathy [40].

However, there are actually more studies showing the negative effects of mTOR signaling on tendons. Oxidative stress activates mTORC1. Tissue metaplasia with heterotopic ossification (HO) and fatty infiltration is observed in tendinopathy [41,42,43,44]. HO is also observed in some cases of acute tendon injury [45]. Aberrant differentiation of TDSCs to non-tenocytes as a result of inflammation has been suggested as the possible cause for tissue metaplasia and poor tenogenesis in tendinopathy [46]. Remarkably, metaplasia in the tendon is associated with the mTOR signaling pathway. The mTOR signaling pathway was activated and mediated fatty infiltration and muscle atrophy after rotator cuff transection and denervation in rats [47,48,49]. Moreover, mechanical overloading also induced activation of mTOR signaling that increased the proliferation and expression of non-tenocyte markers in TDSCs [50]. While low-intensity mechanical loading reduced HO of the tendon through downregulating the mTORC1 signaling pathway, high-intensity mechanical loading activated the mTORC1 pathway and accelerated HO in the rat Achilles tenotomy model and tendon cells in vitro [51]. Inhibition of mTOR, either by rapamycin treatment or gene knockout, inhibited HO in tendon both in vitro and in vivo [52,53]. Besides mTOR signaling, oxidative stress-induced inflammation was also reported to contribute to hypoxia and vascularization associated with the early stages of HO [54].

#### 3.2.4. Metabolic Risk Factors

Obesity and its co-morbidities, including hyperglycemia, diabetes, dyslipidemia and hypercholesterolemia, have been shown to increase the risk of development and poor prognosis of tendinopathy [55,56,57,58]. ROS production, and hence oxidative stress, increases in parallel with adipocyte hypertrophy and adipose tissue hyperplasia, promoting tissue inflammation [59]. The balance of pro- and anti-inflammatory adipokines is perturbed, favoring a systemic low-grade inflammation in obesity. There is compelling evidence that obesity-induced inflammation is causal in the development of many disorders associated with obesity, such as diabetes mellitus, cardiovascular disease and cancer. Recent studies have shown that some of the dysregulated adipokines in obesity also induced HO in tendons via the mTOR pathway. For example, nesfatin-1, an adipokine whose expression is dysregulated in obesity and diabetes, showed increased expression in clinical samples of tendinopathy. It caused HO in tendon tissue via activation of the mTOR pathway and suppression of autophagy both in TDSCs in vitro and in the Achilles tenotomy rat model [52]. The autophagy–lysosomal pathway is a cellular defense mechanism against oxidative stress by selectively removing misfolded or damaged intracellular proteins and organelles, thereby attenuating and reversing the injury caused by oxidative stress [60]. Inhibition of autophagy by oxidative stress-induced mTOR signaling therefore affects tendon homeostasis [30]. Suppression of the mTOR pathway alleviated nesfatin-1-enhanced HO development in rat tendon [52]. Leptin, another adipokine, also promoted osteogenic differentiation of TDSCs in vitro and HO in tendon via the mTORC1 signaling; and rapamycin blocked the process [61].

Obesity induces hyperglycemia and type 2 diabetes. Extracellular glucose has a profound effect on the cellular response to oxidative stress. Oxidative stress at a concentration that is normally anabolic becomes pathological at a high glucose concentration. While low glucose concentration promoted differentiation of H_2_O_2_-stimulated tenocytes, high glucose concentration induced cellular apoptosis [62]. Oxidative stress at a high glucose concentration causes tendon inflammation and degeneration in diabetic tendinopathy. For instance, high glucose concentration up-regulated the mRNA expression of MMP-9 and MMP-13 as well as enzymatic activity of MMP-9 in tendon cells [63]. High glucose concentration was also reported to increase intracellular ROS production via upregulating NOX expression in tenocytes, leading to an increase in mRNA expression of IL-6, type III collagen, MMP-2, tissue inhibitor of matrix metalloproteinase-1 (TIMP-1) and TIMP-2 as well as reducing cell proliferation, compared to cells cultured under normal glucose concentration [64]. In another study, high glucose concentration was shown to increase the mRNA expression of NOX1, IL-6, ROS accumulation and apoptosis of rat tenocytes in vitro, and the effects of high-glucose concentration were reversed by dehydroepiandrosterone (DHEA), an adrenal steroid with antioxidant properties [65]. While there was no difference in the histopathological scores in the diabetic and control tendons, there was higher protein expression of NOX1 and mRNA expression of NOX1, IL-6, MMP-2 and TIMP-2 in the diabetic tendons, compared to that in the control tendons [64]. High NOX level, indicating high inflammatory status, was also observed in diabetic rat tendons in another study [66]. The intraperitoneal injection of DHEA lowered the protein expression of NOX1 and the mRNA expression of NOX1, IL-6, MMP-2, TIMP-2, and type III collagen, along with an increase in the expression of type I collagen in diabetic tendons, compared to the untreated control group [65].

Increased oxidative stress in diabetes can accelerate the formation of advanced glycation end products (AGEs) and collagen cross-linking. H_2_O_2_ was reported to mediate glucose-induced collagen cross-linking in vitro and in vivo [67]. Supplementation of ROS scavengers inhibited glucose-induced oxidative stress, glycation and cross linking of collagen in rat tail tendon ex vivo [68,69]. Intraperitoneal injection of catalase and other antioxidants also suppressed oxidative stress and collagen cross-linking of tendon implanted into diabetic rat peritoneum [67]. Anti-oxidative therapies therefore may protect tendons of diabetic patients.

Hypercholesterolemia is a risk factor for the development of tendinopathy. Cholesterol-induced oxidative stress may contribute to failed tendon healing after tendon injury. In this regard, high cholesterol induced histopathological abnormalities, apoptosis, autophagy and expression FOXO1 in the Achilles tendon of a ApoE knockout hypercholesterolemic mouse model [70]. In the same study, high cholesterol also triggered ROS generation, suppressed proliferation and migration as well as inducing apoptosis and autophagy of rat TDSCs. NAC and FOXO1 inhibitor rescued apoptosis and autophagy induced by cholesterol, the autophagy inhibitor 3-methyladenine (3-MA) enhanced apoptosis, while the apoptosis inhibitor Z-VAD-FMK diminished cholesterol-induced autophagy. This suggested that the effect of cholesterol on TDSCs was mediated by the ROS-activated AKT/FOXO1 pathway and autophagy was a cytoprotective mechanism triggered by cholesterol treatment [70]. In another study by the same group, high cholesterol elevated the ROS level, inhibited the expression of tendon markers in TDSCs and promoted the development of tendinopathy-like changes in the Achilles tendon in the ApoE knockout hypercholesterolemic mouse model via the ROS-activated NF-kB pathway [71]. Both NAC and BAY11-7082 (a broad spectrum IKK (inhibitor of kB kinase)) reversed the inhibitory effects of cholesterol on tendon-related marker expression in TDSCs via blocking NF-kB activation [71].

A high concentration of glutamate, a neuropeptide and excitatory neurotransmitter involved in the transmission of pain and nociception, was reported in areas of painful Achilles, patellar, and lateral epicondylar tendons [72,73,74]. One study reported that glutamate increased intracellular ROS production and apoptosis of rat tendon fibroblasts, supporting the role of glutamate-induced oxidative stress in the pathogenesis of tendinopathy. NAC reduced ROS production and the cytotoxic effects of glutamate on tendon fibroblasts [75].

#### 3.2.5. Drug Exposure

Tendinopathy is a complication of the use of fluoroquinolone antibiotics. The adverse effect of fluoroquinolone antibiotics on tendons is thought to be mediated via oxidative stress. Treatment of human tendon cells with fluoroquinolone antibiotics induced oxidative stress and loss of mitochondrial membrane permeability in human tendon cells. MitoQ, a mitochondria targeted antioxidant, effectively reversed the process [76]. Treatment of rabbit tenocytes with quinolone or fluoroquinolone antibiotics was also reported to increase the intracellular ROS, as well as reducing the redox status, intracellular GSH content and cell viability, compared to the untreated control [77,78]. In a mouse study, the oral administration of a fluoroquinolone antibiotic was shown to induce oxidized damage to type I collagen and altered proteoglycan metabolism in the Achilles tendon, which was prevented by the coadministration of NAC [79]. The results support that mitochondrial damage and subsequent oxidative stress are the potential treatment targets of fluoroquinolone-induced tendinopathy [76].

Anesthetics, steroids and NSAIDs are commonly used for symptomatic relief in tendinopathy. However, they can have detrimental effects on tendon health and predisposed the treated tendon to degenerative changes and rupture after long-term use. Some of them induced tendon damage via ROS generation. For instance, local anesthetic bupivacaine was shown to reduce the viability of rat Achilles tendon cells, and NAC inhibited the process [80]. A single peritendinous injection of bupivacaine increased apoptosis of endotenon cells, increased pro-MMP-9 expression, increased the type III collagen/type I collagen ratio, and transiently reduced the tensile load of the affected Achilles tendon in rats [80]. Dexamethasone induced oxidative stress in tendon cells, along with reduced cell migration, proliferation, types I and type III collagen expression as well as activation of apoptosis [81]. In another study, dexamethasone was also reported to reduce proliferation and tenogenic marker expression in tenocytes in vitro [82]. Figure 1 summarizes the potential sources and mechanisms of oxidative stress-induced tendon damage.

Tendon injury, overuse, metabolic risk factors and drug exposure may increase the expression of inflammatory cytokines, growth factors, which subsequently induce reactive oxidative stress (ROS) production in tendon via nicotinamide adenine dinucleotide phosphate oxidase (NOX) activation. As shown in various pre-clinical studies, the excessive production of ROS generates oxidative stress, which may cause pro-inflammatory response, increase the expression of matrix degrading enzymes, induce cellular apoptosis, reduce autophagy and decrease the viability of tendon cells. It can also reduce the stemness and altered differentiation of tendon-derived stem cells (TDSCs). These cellular changes may further cause fibrosis and adhesion in acute tendon injury as well as matrix degeneration, tissue metaplasia and failed healing in tendinopathy. The mTOR pathway appears to be involved in mediating some effects, such as heterotopic ossification (HO), of oxidative stress on tendon. The imbalance of the redox status and hence, functions of the neighboring cell types, such as myocytes, endothelial cells, macrophages, and adipocytes, may also contribute to tendon pathology. Strategies that scavenger excessive ROS in tendon at different levels of the signaling pathways may be useful for the promotion of tendon healing (MMP—matrix metalloproteinases; Col III—type III collagen; Col I—type I collagen.

## 4. Anti-Oxidative Therapies for the Promotion of Tendon Repair

The association of oxidative stress with cellular damage and tendon pathology suggests that reducing ROS and oxidative stress de facto is a universally desirable treatment strategy. Many studies, though most of them are in vitro and in vivo experiments, have shown the beneficial effects of suppressing oxidative stress on the promotion of tendon healing.

### 4.1. Clinical Trials

The effects of antioxidant supplements on the promotion of tendon repair have been examined in several clinical studies. In a randomized controlled trial of patients with rotator cuff repair, the use of a supplement containing antioxidants (arginine L-alpha-ketoglutarate, methylsulfonylmethane, and bromelain) and hydrolyzed type I collagen for 3 months after repair decreased shoulder pain and improved repair integrity but had no effect on the objective functional outcomes [83]. In another study, insertional Achilles tendinopathy patients receiving daily dietary supplement containing arginine, Vinitrox (Bio Serae Laboratories SAS, Bram, France), collagen, methyl–sulfonyl–methane, vitamin C, and bromelain (the last three ingredients are antioxidants) in addition to extracorporeal shockwave therapy (ESWT) showed better outcomes (Ankle-Hindfoot Scale score, Roles and Maudsley score and lower pain level and oximetry value) compared to the placebo group receiving only ESWT [84]. In yet another double-blinded randomized placebo-controlled trial investigating the effect of the daily supplementation of essential fatty acids (eicosapentaenoic acid—EPA, docosa-hexaenoic acid—DHA, gamma–linolenic acid—GLA) and antioxidants (vitamins C and E, selenium, zinc, vitamin A, and vitamin B6) combined with therapeutic ultrasound for the treatment tendinopathy in recreational athletes, patients in the treatment group reported a significant reduction in pain and an increase in sport-specific activity compared to the placebo group. One limitation of this study is an uneven distribution of tendinopathy locations in the two groups, making the interpretation of results difficult [85]. In all three controlled trials, the supplement contained a mixture of different ingredients. The exact effects of antioxidants and the specific antioxidant on the treatment of tendinopathy remain to be seen. One randomized controlled trial failed to show a difference in active mobilities, constant score, subjective shoulder value and ultrasound classification in the vitamin C supplementation group compared to the no supplementation group in patients undergoing rotator cuff repair, despite the insignificant lower non-healing rate in the treatment group at 6 months post-surgery [86].

While there was an animal study showing the anti-oxidative and healing effects of low-intensity laser therapy (LLLT) on tendon [87], one single session of high-intensity laser therapy (HILT), which is speculated to show similar biochemical effect to LLLT, was ineffective in returning the oxidant–antioxidant equilibrium in young amateur athletes with torn or pulled tendons of the ankle or the knee joint [88]. However, it enhanced the stability of lysosomal membranes as demonstrated by lower serum activities of acid phosphatase (AcP) and arylsulfatase (ASA) (both lysosomal enzymes) in the injured participants, compared to the uninjured participants 30 min after HILT treatment [88]. This study has several limitations. First, the longer-term treatment effects of HILT were not examined. The choice of healthy participants, but not untreated injured athletes, as the controls is also a concern. Moreover, there were no difference in the redox parameters and lysosomal hydrolases between the healthy and injured athletes at baseline prior to HILT, questioning the choices of outcomes in this study [88]. At present, there is a lack of high-quality evidence of the effect and underlying mechanism of photobiomodulation for the treatment of tendinopathy [89].

### 4.2. Pre-Clinical Studies

#### 4.2.1. mTOR Pathway Inhibitors

The mTOR signaling pathway is involved in oxidative stress-induced HO and appears to be a viable therapeutic target for the treatment of HO in tendon. The suppression of the mTOR pathway and activation of autophagy with rapamycin were reported to alleviate HO development in a rat Achilles tenotomy HO model [51,52] and in tendon cells [51]. Moreover, nesfatin-1 accelerated HO in tendon, and the effects were rescued by rapamycin treatment [52]. The effect of rapamycin in the prevention of HO was further confirmed by its targeted delivery to the pathological collagen by collagen hybrid peptide-modified poly (lactic–coglycolic acid) (CHP-PLGA) nanoparticles. The rapamycin-loaded CHP-PLGA nanoparticles specifically bound to the pathological tendon and strongly suppressed HO progression in a mouse collagenase-induced tendon injury model [53]. Furthermore, rapamycin was reported to inhibit the mechanical loading-induced activation of mTOR, proliferation and expression of non-tenocyte markers in rat patellar TDSCs [50]. Additionally, daily injection of rapamycin suppressed overuse-induced degenerative changes in mouse tendon [50]. In addition to its anti-mineralization effect on injured or overloaded tendon, rapamycin was shown to inhibit fatty infiltration in the rat rotator cuff injury model [48]. Rapamycin decreased intracellular and mitochondrial ROS accumulation and activated cytoprotective autophagy in hTDSCs, thereby protecting hTDSCs against H_2_O_2_-induced loss of self-renewal capacity and stemness [30]. Umbilical cord stem-cell-derived exosomes were also reported to promote tendon regeneration via activation of mTOR signaling, both in TDSCs in vitro and in the rat Achilles tendon injury model [90]. 

#### 4.2.2. Hyaluronic Acid

Hyaluronic acid (HA) is a well-known anti-fibrotic and anti-inflammatory agent that reduces scar formation and adhesion during tissue healing. It prevented post-operative tendon adhesion in both animal and human studies and has been used in many degenerative conditions, such as osteoarthritis [91]. Direct tendon injection of HA maintained the architecture, reduced microtearing and decreased apoptosis of tendon in rat tendinopathy models [92,93]. The application of HA to the repaired site was reported to counteract inflammation, alleviate pain and improve tendon-to-bone healing in a rabbit rotator cuff repair model [94,95]. It was reported that HA reduced ROS production in mechanical-stress-loaded bovine cartilage [96]. A study reported that HA protected H_2_O_2_-treated human tenocytes from cytotoxicity, upregulated the expression of CD44 (an HA receptor), enhanced catalase activity recovery, reduced Nrf2 protein expression and increased autophagy [97]. Treatment of H_2_O_2_-damaged human tenocytes with methylsulfonylmethane (a naturally occurring antioxidant) conjugated to hyaluronic acid protected the cells from oxidative-stress-induced cytotoxicity, and reduced iNOS and PGE_2_ production [98].

#### 4.2.3. Natural Antioxidants from Plants 

Flavonoids are a large group of natural compounds possessing high anti-oxidative activities. Administration of quercetin, a flavonoid, was reported to show protective effect against collagenase-induced tendinopathy in rats via suppressing oxidative stress, inflammation, apoptosis, autophagy and matrix degeneration [22]. The suppression of autophagy marker beclin 1 by quercetin in this study was inconsistent with the previous studies about the cytoprotective effect of autophagy on tendon; further research is needed. Similarly, quercetin was also reported to suppress oxidative stress and reduce tendon adhesion after transection and repair in rats with higher levels of SOD and glutathione peroxidase (GPx) and lower levels of MDA [99].

Oral administration of curcumin [100], a flavonoid polyphenol, was shown to reduce oxidative stress and lipid peroxidation products in serum and collagen cross linking in the tail tendon of diabetic rats. A smart oxidative stress-responsive electronspun polyester membrane acting both as a physical barrier and as reservoir for the slow release of curcumin/celecoxib was shown to prevent peritendinous adhesion in a rat Achilles tendon transection and repair model [101].

Proanthocyanidins and anthocyanins are the main phenolic compounds of grape skin and are strong antioxidants of the flavonoid family. Proanthocyanidins was reported to improve the viability of H_2_O_2_-exposed TDSCs via upregulating the protective Nrf2 signaling pathway [102]. In other studies, anthocyanins were also reported to dose-dependently inhibit intracellular ROS formation and apoptosis of tendon cells exposed to H_2_O_2_ [103,104,105] showed that H_2_O_2_ induced autophagic cell death in tendon cells, and cyanidin (a type of anthocyanins) inhibited the effect of H_2_O_2_.

Apocynin is a vanilloid compound, naturally found and obtained from the root extract of the medicinal herb *Picrorhiza kurroa*. Apocynin is a known inhibitor of NOX activity and exerts its activity by blocking the formation of the NOX complex. Apocynin was demonstrated to promote cell proliferation and suppress ROS production, inflammatory cytokine expression and cell death in rat tenocytes exposed to a high concentration of glucose via NOX inhibition and hence may be useful as a drug for the treatment of diabetic tendinopathy [106].

The catechin epigallocatechin gallate (EGCG) is the main flavonoid compound of green tea. Both oxidized hyaluronic acid/adipic acid dihydrazide hydrogel (HA hydrogel) and EGCG-loaded HA hydrogel suppressed the type III collagen/type I collagen gene ratio in human tendon cells after 8% intensive mechanical loading (8 h/day for 3 days) [107]. The gene expression of non-tenocyte markers after treatment with HA hydrogel and EGCG-loaded HA hydrogel was reduced, but the differences were not statistically significant. However, in another study, the supplementation with antioxidants, EGCG and piracetam, respectively, significantly suppressed the mRNA expression of non-tenocyte markers and type III collagen/type I collagen ratio in intensively loaded tendon cells [108]. In animals, a single injection of EGCG-loaded hydrogel one day after collagenase injection improved the histological score compared to the saline injection group at day 14 post-injury in the collagenase-induced Achilles tendinopathy model. The mRNA expression of type III collagen/type I collagen ratio decreased, and the mRNA expression of non-tenocyte markers decreased in the tendon in both the HA hydrogel group and the EGCG-loaded HA hydrogel group, but the differences were statistically significant only for the expression of the type III collagen/type I collagen ratio in both groups and PPARg (an adipogenic marker) in the EGCG-loaded HA hydrogel group [108].

Eugenol is one of the key volatile constituents of cinnamon and clove oils, with an anti-oxidative property. Extracellular-vesicles-derived bone marrow stromal cells (BMSCs) pretreated with eugenol were shown to inhibit ROS accumulation and apoptosis as well as promoting catalase and SOD1 expression, viability, proliferation, and tenogenic differentiation of H_2_O_2_-treated TDSCs, compared to the untreated BMSC-EV group via the Nrf2/HO-1 pathway [20]. The transplantation of eugenol-BMSC-EV-pretreated TDSCs showed significantly improved tenogenesis and matrix regeneration compared to the untreated TDSCs during tendon healing [20]. The mechanism, including the anti-oxidative effect, of eugenol-BMSC-EV-pretreated TDSC was not examined in the animal model. Further research is required.

Berberine is an antioxidant found in some plants, such as European barberry, goldenseal, goldthread, Oregon grape, phellodendron, and tree turmeric. Treatment with berberine reversed the effects of dexamethasone on tendon cells [81]. Oral administration of chloroform extract of *Azadirachta indica* and methanolic extract of *Cuminum cyminum* was shown to reduce oxidative stress and collagen cross linking in rat tail tendon in streptozotocin-induced diabetic rats [109,110].

#### 4.2.4. Ascorbic Acid/Vitamin C

Ascorbic acid is an antioxidant that promotes collagen biosynthesis and prevents free radical formation. There is evidence showing that ascorbic acid may protect tendon tissue from oxidative stress. For instance, a low dose of ascorbic acid increased cell proliferation, viability and migration of TDSCs [29]. The administration of ascorbic acid to the SOD−/− degenerative supraspinatus enthesis mouse model attenuated oxidative stress-induced histopathological changes in the tendon [111]. In addition, local injection of ascorbic acid increased tissue GSH level, reduced gliding resistance, fibrotic size and peritendinous adhesion, compared to the saline injection group in a chicken flexor digitorum profundus tendon injury model [34]. Co-treatment with ascorbic acid was shown to reduce the cytotoxic effects of bupivacaine (an analgesic) and ketorolac tromethamine (a NSAID) on human tenocytes isolated from a torn supraspinatus tendon, suggesting that ascorbic acid might be useful to mitigate the side effects of analgesics and NSAIDs for treating patients with tendinopathy [112].

#### 4.2.5. Vitamin D

Vitamin D was demonstrated to suppress ROS generation as well as reversing the anti-proliferative and anti-tenogenic effects of dexamethasone on tenocytes, suggesting that it might have a beneficial effect on dexamethasone-induced tendon injury [82].

#### 4.2.6. Vitamin E

Local administration of Trolox (6-hydroxy-2,5,7,8-tetramethylchroman-2-carboxylic acid), a water-soluble analog of vitamin E, reduced tendon gliding resistance, fibrotic mass and adhesion in a chicken flexor digitorum profundus tendon injury model [113].

#### 4.2.7. Protection of Mitochondria

Hypoxia is frequently associated with oxidative stress and subsequent inflammation and apoptosis of the tendon. Mitochondria regulate cellular oxidative stress and apoptosis. Indeed, mitochondrial dysfunction was observed in a mouse model of supraspinatus tendinopathy induced by placing a microsurgical clip in the subacromial space [114]. The mitochondrial number, expression of mitochondrial-related genes and SOD activity in the supraspinatus tendon decreased at week 4 after clip placement but increased following clip removal. The authors concluded that mitochondrial protection might protect tendon and promote tendon healing. Subsequently in the same year, one research group transplanted exogeneous intact mitochondria isolated from mesenchymal stromal cells derived from umbilical cord (UC-MSCs) to TNF-α-treated tenocytes and successfully attenuated oxidative stress, promoted mitochondrial functional recovery, restored the expression of tenogenic markers, inhibited apoptosis, and reduced the expression of pro-inflammatory markers and pNF-kB in damaged tenocytes [115]. Similarly, local injection of mitochondria isolated from L6 myoblasts to the Achilles tendon 2 weeks after collagenase-induced injury reduced the expression of apoptosis and pro-inflammatory markers, pNF-kB, MMP1 and increased collagen content 2 weeks after injection [115]. Pretreatment of Achilles tenocytes with Alda-1, a selective activator of mitochondrial aldehyde dehydrogenase 2 (ALDH2), which alleviates oxidative stress and ER stress, attenuated H_2_O_2_-induced cell death, oxidative stress, mitochondrial membrane depolarization and apoptosis [23]. Alda-1 treatment also reduced the severity of H_2_O_2_-indcued Achilles tendinopathy in vivo by reducing apoptotic cell death and the expression of inflammatory cytokines IL-1β and TNF-α [23].

#### 4.2.8. Others

Melatonin is a powerful antioxidant. Melatonin-loaded polycaprolactone/sodium alginate scaffold was reported to activate the Nrf2/HO-1 signaling pathway and inhibited ROS production and macrophage infiltration, thereby promoting tendon repair, after transplantation to the Achilles tendon injury site [116]. The administration of melatonin was also shown to reduce the plasma total antioxidant status (TAS), total oxidant status (TOS), TOS/TAS ratio, iNOS, VEGF post-treatment compared to baseline in a supraspinatus overuse tendinopathy rat model [36]. However, no direct comparison with the control group was done in this study and so the conclusion should be interpreted with caution. 

BPC-157 is a synthetic pentadecapeptide that has been investigated for inflammatory bowel disease and soft tissue healing. Both intraperitoneal and oral administration of BPC 157 was reported to restore injured myotendinous junction in rats with reduced inflammatory infiltrate, oxidative stress and NO-levels [117]. However, as of 1 January 2022, this compound is not approved for human clinical use and is prohibited under the World Anti-Doping Agency (WADA) Prohibited List. 

CO-releasing molecules (DCH-CORMs) were reported to protect human tenocytes from oxidative stress. For instance, dicobalt(0)hexacarbonyl-CO-releasing molecules (DCH-CORMs) were reported to reduce H_2_O_2_-induced oxidative stress and PGE_2_ secretion via CO release and COX-2 inhibition in human tenocytes [118]. Moreover, two carbonic anhydrases inhibitors–CO-releasing molecules (CAIs-CORMs) were shown to inhibit oxidative stress-induced cytotoxicity, enhance viability and augment proliferation of H_2_O_2_-stimulated human tenocyte, compared to treating cells with Meloxicam (a NSAID) [119]. One CAI (compound **2**) was shown to exert its action via inhibiting NF-kB translocation and downregulating iNOS, while the other CAI (compound **7**) was more effective in increasing type I collagen deposition [119].

Anethole dithiolethione, a well-known antioxidant and glutathione inducer, was reported to protect rabbit tendon cells from fluoroquinolone-induced oxidative stress and cytotoxicity [120]. 

The administration of hydrogen water was shown to reduce tendon adhesion after tendon repair in rats, with a concomitant increase in SOD activity and GSH contents and reduction in MDA, compared with the normal saline group [32].

The role of PRP on tendon healing remains controversial. In an early study, platelet-released growth factors enhanced tenocyte growth, migration and the Nrf2-ARE (antioxidant response element) pathway, suggesting that PRP releasate has an anti-oxidative effect on tenocytes [121]. Another study also reported that activated human PRP reduced ROS production and the mRNA expression of inflammatory markers, MMP-3 and thrombospondin motifs-4 (ADAMTS-4) in human tendon cells exposed to IL-1β [122]. While triamcinolone (a corticosteroid) similarly reduced ROS production and the mRNA expression of inflammatory markers in the inflamed human tendon cells, it reduced cell viability and has no effects on the rescue of matrix degeneration [122]. The supplementation of PRP could mitigate the negative effects of triamcinolone, suggesting the use of activated PRP alone or combined treatment of PRP and triamcinolone to relieve symptoms of rotator cuff tendinopathy [122]. However, a more recent study reported that PRP induced proinflammatory signaling pathways and oxidative stress in tendon fibroblasts [123]. The discrepancy may be due to different PRP preparation. In fact, there is a discrepancy in clinical outcomes and effects after PRP treatment in tendinopathic patients, probably due to different protocols of PRP preparation despite its positive effects in cell culture and animal models [123]. Further studies are required to understand the effects of PRP on oxidative stress and functions of tendon cells.

Fillipin et al. [87] showed that LLLT protected against oxidative stress and fibrosis of traumatized Achilles tendons in rats. They showed that the concentration of thiobarbituric acid reactive substance (TBARS, lipid–peroxidation end products), chemiluminescence induced by hydroperoxides and collagen concentration in damaged tendons were significantly lower, while the SOD activity was higher after LLLT treatment. The levels of inflammation and fibrosis were also reduced after LLLT treatment. Further research is needed to show the anti-oxidative effect of LLLT on tendon.

Compared with traditional suture repair, the rats additionally treated with a combination of upconversion nanoparticles and photochemical tissue bonding technology had better effects on Achilles tendon repair in rats [124]. The benefits might be related to the transient, but not excessive, generation of ROS in the early stage of tendon healing [124]. Similarly, photochemical tissue bonding treatment for the right treatment time was also reported to promote the proliferation and migration of injured tenocytes through increasing ROS production, possibly also not at the excessive level [125].

## 5. Future Research Directions

While numerous in vitro and in vivo studies have demonstrated the role of oxidative stress in mediating the damaging effect of tendon injury, overuse, metabolic risk factors, and drug exposure on the tendon, supportive clinical evidence is lacking. Further studies should examine the source and roles of oxidative stress in mediating their damaging effects on tendons in the clinical setting, which may also contribute to the identification of new diagnostic and prognostic biomarkers. 

Other intrinsic and extrinsic factors, such as age and smoking, may also induce systematic oxidative stress and affect tendon health. Further research on the effects of these ROS sources on tendon is needed. Smoking is an important risk factor for the development of chronic tendinopathy [126,127]. Nicotine from cigarette smoke is well known to induce mitochondrial ROS production and oxidative stress that triggers a generalized inflammation associated with cytokine release, adhesion of inflammatory cells and disruption of tissue, such as endothelium and renal tubules [128,129]. In tendons, nicotine was reported to downregulate the expression of MMP and TIMP expression in tenocytes under loading condition, suggesting that it may affect the ECM remodeling of tendon, predisposing it to rupture during loading [130]. However, there has been no mechanistic study examining the direct effects of cigarette smoke or nicotine on oxidative stress of tendon cells, making it unclear about its underlying pathogenic mechanism on tendinopathy. Further research in this area is warranted.

Tendinopathy is a broad spectrum of tendon disorders with different pathological changes. Fibrosis and adhesion after acute tendon injuries also vary with anatomical locations. The precise mechanisms of oxidative stress in mediating tendon damage at different locations merit further investigation. Research on the molecular mechanisms including activators of oxidative stress, cell types producing oxidative stress and the specific ROS types in different tendon injuries and disorders would shed light on the discovery of new treatment targets. Tendons can also be indirectly influenced by changes in ROS metabolism in other tissues and cells, such as exercising muscles, vascular tissue and adipose tissue. Further research should be done to investigate the influence of redox homeostasis of other tissues on tendon. 

Antioxidants are generally demonstrated to have beneficial effects on the viability and tenogenic differentiation of tendon cells as well as HO, fatty tissue infiltration, tendon adhesion and tendon repair in animal models. Some potential treatments have limited supporting data in vitro and in vivo. Further research to demonstrate their efficacies in reversing oxidative stress and promoting tendon healing at different dosages, timing of application, duration, and animal models are needed before clinical trial of the promising candidates. Further research to identify novel antioxidants for the promotion of tendon healing is needed. 

There is a paucity of clinical studies of anti-oxidative therapies for the promotion of tendon healing. The few clinical studies on the effects of nutritional supplements for tendon repair involve a mixture of ingredients, and the study design makes it impossible to dissect the effects of antioxidants and specific antioxidants on tendon repair. There is a clear need for well-planned randomized controlled trials to investigate the efficacy of a single antioxidant in the management of tendinopathy.

## 6. Conclusions

There is vast pre-clinical and some clinical evidence that oxidative stress due to excessive ROS production might contribute to the pathogenesis of acute tendon injury and degenerative tendinopathy. Pre-clinical data have shown that tendon injury, overuse, metabolic risk factors and drug exposure are potential sources of ROS in tendon disorders. In vitro and in vivo studies have shown that anti-oxidative therapies might be useful to suppress oxidative stress and promote tendon healing. However, clinical evidence about their efficacies is lacking. Further research to understand the molecular mechanisms of oxidative stress-induced tendon damage and develop novel anti-oxidative therapies for the promotion of tendon healing is required.

## Figures and Tables

**Figure 1 ijms-23-03571-f001:**
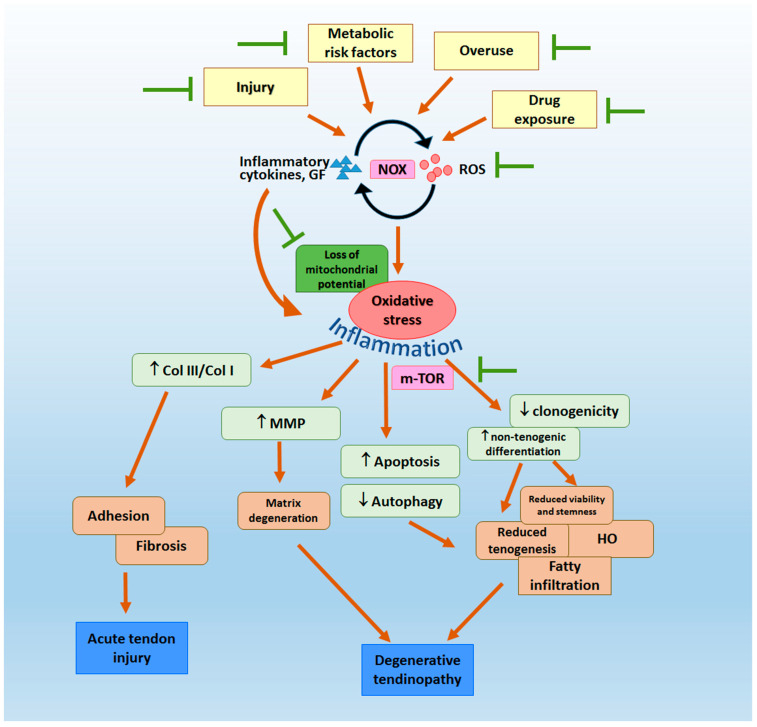
Schematic diagram summarizing the potential sources and mechanisms of oxidative stress-induced tendon damages.

## Data Availability

Not applicable.

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
