# Peer review of "Roles of Oxidative Stress in Acute Tendon Injury and Degenerative Tendinopathy—A Target for Intervention"

_ijms, 2022, doi:10.3390/ijms23073571_

Round 1

Reviewer 1 Report

The review:  “Roles of Oxidative Stress in Acute Tendon Injury and Degenerative Tendinopathy – a Target for Intervention”  is very interesting as it is a wide scientific overview.

However, I would like to invite the authors  to clarify some points:

  1. The paragraph of the conclusions appears too long, the authors could rewrite it by better reorganizing the concepts.
  2. Page 9: in the context of HA application for OA management the authours should cite this scientific work: Vassallo V, Stellavato A, Cimini D, Pirozzi AVA, Alfano A, Cammarota M, Balato G, D'Addona A, Ruosi C, Schiraldi C. Unsulfated biotechnological chondroitin by itself as well as in combination with high molecular weight hyaluronan improves the inflammation profile in osteoarthritis in vitro model. J Cell Biochem. 2021 May 31;122(9):1021–36. doi: 10.1002/jcb.29907. Epub ahead of print. PMID: 34056757; PMCID: PMC8453819.

Author Response

Reviewer 1:

The review:  “Roles of Oxidative Stress in Acute Tendon Injury and Degenerative Tendinopathy – a Target for Intervention”  is very interesting as it is a wide scientific overview.

However, I would like to invite the authors  to clarify some points:

The paragraph of the conclusions appears too long, the authors could rewrite it by better reorganizing the concepts.

Response: Thank you very much for your comment. We have rewritten the conclusion to be made more precise and concise.

Page 9: in the context of HA application for OA management the authors should cite this scientific work: Vassallo V, Stellavato A, Cimini D, Pirozzi AVA, Alfano A, Cammarota M, Balato G, D'Addona A, Ruosi C, Schiraldi C. Unsulfated biotechnological chondroitin by itself as well as in combination with high molecular weight hyaluronan improves the inflammation profile in osteoarthritis in vitro model. J Cell Biochem. 2021 May 31;122(9):1021–36. doi: 10.1002/jcb.29907. Epub ahead of print. PMID: 34056757; PMCID: PMC8453819.

Response: Thank you very much for your comments. The reference is added.

Reviewer 2 Report

Dear Authors,

This is a very interesting review article. The current concepts on pathogenesis and treatment with future research direction are discussed.

Nevertheless, I have some suggestions to improve the paper:

  1. I suggest changing the place of the lines 72-83 and putting them in the abstract.
  2. I suggest adding some information about the effect of smoking on the rotator cuff tendons in the 3.2. section
  3. Please add a bit more information about the effect of PRP on tendons in the 2.8. section (e.g. Neovascularization in Meniscus and Tendon Pathology as a Potential Mechanism in Regenerative Therapies: Special Reference to Platelet-Rich Plasma Treatment Applied Sciences 11 (18), 8310)

Author Response

Reviewer 2

This is a very interesting review article. The current concepts on pathogenesis and treatment with future research direction are discussed.

Nevertheless, I have some suggestions to improve the paper:

I suggest changing the place of the lines 72-83 and putting them in the abstract.

Response: Thank you very much for your comment. We have put line 72-83 in the abstract and shortened the paragraph of the study aim in the main text. Since this is a systematic review of the literature, we believe that it is necessary to give details about how we searched the articles in the main text. Therefore, we prefer to keep the search strategy in the main text, if the reviewer agrees.

I suggest adding some information about the effect of smoking on the rotator cuff tendons in 3.2. section

Response:

Thank you very much for your comment. Smoking is an important risk factor for the development of chronic tendinopathy. Nicotine from cigarette smoke is well known to induce mitochondrial ROS production and oxidative stress that triggers a generalized inflammation associated with cytokine release, adhesion of inflammatory cells, and disruption of tissue such as endothelium and renal tubules.  In tendons, nicotine was reported to downregulate the expression of MMP and TIMP expression in tenocytes under loading conditions, suggesting that it may affect ECM remodeling of the tendon, predisposing it to rupture during loading. However, there has been no mechanistic study examining the direct effects of cigarette smoke or nicotine on oxidative stress of tendon cells, making it unclear about its underlying pathogenic mechanism on tendinopathy. Further research in this area is warranted. As there has been no report on the direct effect of smoke on ROS generation in tendons, we include the information in the section on “Future research direction” instead of section 3.2, if the reviewer agrees.

Please add a bit more information about the effect of PRP on tendons in 2.8. section (e.g. Neovascularization in Meniscus and Tendon Pathology as a Potential Mechanism in Regenerative Therapies: Special Reference to Platelet-Rich Plasma Treatment Applied Sciences 11 (18), 8310)

Response: Thank you very much for your comment. We have added more information about PRP on tendons and added the reference.

Round 2

Reviewer 2 Report

I accept in present form.